# Mortality and Costs of Cardiac Implantable Electronic Device (CIED) Infections According to the Therapeutic Approach: A Single-Center Cohort Study

**DOI:** 10.3390/microorganisms12030537

**Published:** 2024-03-07

**Authors:** Encarnación Gutiérrez-Carretero, Eduardo Arana-Rueda, Antonio Ortiz-Carrellán, Alonso Pedrote-Martínez, Mariano García-de-la-Borbolla, Arístides De Alarcón

**Affiliations:** 1Cardiac Surgery Service, Centro de Investigación Biomédica en Red en Enfermedades Cardiovasculares (CIBERCV), Institute of Biomedicine of Seville (IBiS), Centro Superior de Investigaciones Científicas (CSIC), University Hospital Virgen del Rocío, University of Seville, 41013 Seville, Spain; gutierrez.encarnita@gmail.com; 2Electrophysiology and Arrhythmia Unit, Department of Cardiologý, University Hospital Virgen del Rocío, 41013 Seville, Spain; eduaru@hotmail.com (E.A.-R.); pedroteal@hotmail.com (A.P.-M.); 3Echocardiography Unit, Department of Cardiologý, University Hospital Virgen del Rocío, 41013 Seville, Spain; aortizcarrellan@yahoo.es; 4Cardiac Surgery Service, University Hospital Virgen del Rocio, 41013 Sevilla, Spain; marianogbf@gmail.com; 5Clinical Unit of Infectious Diseases, Microbiology and Parasitology, Bacterial Resistance and Antimicrobials Research Group, Centro de Investigación Biomédica en Red en Enfermedades Infecciosas (CIBERINFEC), Institute of Biomedicine of Seville (IBiS), ESCMID Collaborative Centre, Centro Superior de Investigaciones Científicas (CSIC), University Hospital Virgen del Rocio, University of Seville, 41013 Seville, Spain

**Keywords:** cardiac, implantable, electrostimulation, devices, cost, infection

## Abstract

**Background:** Cardiac device infections are serious adverse events associated with considerable morbidity and mortality, significant costs, and increased healthcare utilization. The aim of this study is to calculate the costs of treatment of cardiac implantable electrostimulation device (CIED)-related infections for different types of infection (local or systemic) and therapeutic approaches. **Patients and Methods**: Single-center cohort (1985–2018). The costs of the CIED-related infections were analyzed according to initial treatment (antimicrobial treatment exclusively, local approach, or transvenous lead extraction (TLE)). Total costs (including those for hospitalization stay, drugs, extraction material, and newly implanted devices) were assigned to each case until its final resolution. **Results**: A total of 380 cases (233 local and 147 systemic infections) were analyzed. The average cost of systemic infection was EUR 34,086, mainly due to hospitalization (78.5%; mean: 24 ± 14 days), with a mortality rate of 10.8%. Local infection had a mortality rate of 2.5% (mainly related to the extraction procedure) and an average cost of EUR 21,790, which was higher in patients with resynchronization therapy devices and defibrillators (46% of total costs). Surgical procedures limited to the pocket for local infections resulted in a high rate of recurrence (87%), evolved to systemic infections in 48 patients, and had a higher cost compared to TLE (EUR 42,978 vs. EUR 24,699; *p* < 0.01). **Conclusions**: The costs of treating CIED-related infections are high and mainly related to the type of treatment and length of hospitalization. Complete device removal is always the most effective approach and is a cost-saving strategy.

## 1. Introduction

The use of cardiac implantable electronic devices (CIEDs) has increased due to changing population demographics (an increased number of aged and fragile people with greater comorbidities) and the variety of CIEDs available, which has evolved from fixed rate single-chamber pacemakers to variable rate multichambered pacemakers with the capabilities for cardioversion and defibrillation (implantable cardioverter-defibrillators (ICDs)) and/or cardiac resynchronization therapy (CRT) [1,2]. This trend has involved higher costs and an unexpected increase in device infections in recent years [3,4].

Infections related to cardiac implantable electronic devices (CIEDs) are associated with increased patient morbidity and mortality, representing a significant economic burden for the healthcare system. In 2011, Sohail, M.R. et al. [5] published the first risk-adjusted, device type-specific estimates of mortality, length of stay, and costs associated with CIED infections in the USA. In that study, mortality and costs were very high and it was also found that intensive care and pharmacy services accounted for more than half of the total costs of infections. European studies have reported similar conclusions, although lower costs [6].

The consensus is that complete device removal is the best treatment to ensure complete cure, especially for systemic infections where the intravascular component is compromised [7,8,9]. However, complete extraction by transvenous lead extraction (TLE) of a device that has been implanted for years requires experience and expertise and must be performed in a safe environment [10,11], factors which are not always readily available in all hospitals. Moreover, even in the most experienced referral centers, the rate of major complications associated with this procedure remains around 2–4%, and sometimes open-heart surgery may be required [12]. Therefore, chronic suppressive antimicrobial therapy for systemic infections has been recommended for elderly and frail patients who may not survive open-heart surgery [13,14]. For local infections, which are often characterized by their low aggressiveness, local approaches, such as pocket debridement, in situ generator replacement, or sometimes generator removal and contralateral system insertion leaving the old cables in place, are common practices [15].

However, in our own experience [16], the rate of clinical failure with these local approaches is unacceptable, resulting in a plethora of new problems [17,18] and thus increasing the costs of the procedures. The aim of this study was to analyze the costs of these CIED-related infections, considering the efficacy of the different techniques and the costs associated with each approach.

## 2. Patients and Methods

### 2.1. Study Population and Period

Patients with either local or systemic CIED-related infections who were consecutively treated between 1985 and 2018 at our center, a tertiary hospital in the south of Spain, which conducts more than 600 implantations of CIEDs every year and also acts as a referral center for CIED infections at other hospitals in the region.

Data collection, microbiological procedures, antimicrobial treatment, surgical procedures, and reimplantation of the electrostimulation device are described in detail in the Appendix A.

### 2.2. Types of Infection

We classified CIED infections depending on their location according to international consensus guidelines [7,8,9] and, for a better understanding of their origin, depending on their time of onset, as follows:

#### 2.2.1. Depending on Their Location

-Local infection: When systemic symptoms were not present (fever, shock, embolisms, or remote infectious complications), blood cultures were negative, and there were signs in the generator pocket such as pain, swelling, erythema, and purulent collection objectified by dehiscence of the wound or needle puncture or exudation by chronic fistula. The cases of local or partial generator and/or cable extrusion were always considered infected since they are contaminated with skin flora. The cases of pre-decubitus without skin break were considered local infections if during surgery, purulent material was observed in the pocket and/or leads or, otherwise, if the cultures from the extracted leads were positive.-Systemic infection: When the patient had fever, shock, embolisms, or remote infectious complications (such as spondylitis), and the blood cultures were positive. Cases with negative blood cultures required the presence of vegetation in the leads or right-sided cardiac valvular structures, or positive cultures from the distal end of the extracted leads. Cases with complicated local infections (local symptoms with positive blood cultures) were considered as systemic infections.

#### 2.2.2. Depending on the Time of Onset 

-Acute infections: those appearing before 1 month after device implantation or manipulation.-Delayed infections: those appearing between 2 and 12 months after implantation or manipulation;-Late infections: those appearing after 12 months.

### 2.3. Type of Infection Assignment

Infections were assigned to local or systemic type depending on their presentation (*initial classification*) that motivated the initial therapeutic approach. However, a considerable number of local infections progressed to systemic infections throughout the entire process, which had an impact on its evolution and final mortality. For this reason, we have also considered in the analysis its definitive diagnosis, at the end of the process (*final classification*). The costs and final evolution of systemic infections, therefore, include these originally local infections that evolved to systemic, and those that presented as systemic from the beginning.

### 2.4. Therapeutic Approach

For the analysis, the following options were considered:Exclusive antimicrobial treatment: if no local or complete surgery (TLE or open-heart procedure) was performed.Local surgical approach: debridement of the generator pocket, with or without new generator replacement, or generator extraction and placement of an alternative system, leaving the old leads in place, were included in this option.Complete extraction of the system: either by TLE or open-heart surgery. Cases of lead fracture with residual intravascular material, regardless of size, were considered as incomplete extraction.

### 2.5. Definitions in Outcome

Mortality due to CIED infection in this study was considered to be in-hospital mortality for whatever reason: septic shock from the infection, complications due to the technique (i.e., vein rupture or cardiac tamponade) or the hospitalization (i.e., a nosocomial pneumonia after the extraction) or a worsening of the previous heart disease, such as ventricular failure. Mortality of unknown origin until one month after the discharge was also considered to be attributable mortality.

Non-related mortality was considered to be mortality that occurred due to other reasons apart from the infection during the entire follow-up period (late mortality).

The criteria for sepsis and septic shock used varied according to the current guidelines during the period of the study [19,20,21].

### 2.6. Cost Analysis

In each episode, the costs were grouped into three expenditure items:-The stay cost was determined by the days of total registered days (in one or more centers), assigning the price of the stay/day in either conventional hospitalization or intensive care units. In addition, the costs of the used antimicrobial treatment, which in many cases was extended beyond the stay days, and those derived from outpatient visits due to the process were also included.-The costs related to the material used in the extraction process were also included in the analysis. However, no other material cost used in the operation room was computed.-Finally, the costs of the new implanted devices after the extraction of the old device were also included. Since the cost of these devices has varied over time, the average costs for each device in the year 2018 were included in this study.

### 2.7. Follow-Up

Patients in our hospital were followed for one year in an infectious disease clinic and then for life in the electrophysiology unit. For patients referred from other centers, the minimal follow-up in our hospital consisted of a three-month period of blood culture control for systemic infections and maintaining contact with the reference hospital for one year.

The persistence of the original signs and symptoms with continued isolation of the same microorganism or their recurrence after a specific therapeutic approach was defined as treatment failure.

If signs of infection due to the same micro-organism isolated from the originally extracted device were detected in the newly implanted device during the first year of follow-up, this episode was considered a relapse. However, if a different microorganism was detected during that year, the event was recorded as an early reinfection. The episodes of relapse and early reinfection were registered as part of the same process, adding up the costs until the final resolution. If the reinfection with a different microorganism occurred after this one-year period, it was considered a late reinfection and registered as a separate episode.

### 2.8. Risk of Infection

For evaluating the risk of infection (before this episode) in each patient, we used the PADIT score. The PADIT risk score was developed by Birnie et al. [22], using the population of the PADIT trial [23] and is currently the only risk score not derived from retrospective analysis. Five easy-to-access, independent predictors were recognized, namely prior procedures, age, depressed renal function, immunocompromised, and procedure type, giving a score ranging from 0 to 15 points. This classified patients into low (0 to 4), intermediate (5 to 6), and high (≥7) risk, with rates of hospitalization for infection of 0.51%, 1.42%, and 3.41%, respectively. In this study, patients were considered high-risk subjects if they had a PADIT score of >6 or if they had a previous infection registered.

### 2.9. Statistical Analysis

While continuous variables were represented as either mean ± standard deviation (SD) or median and interquartile range (IQR), categorical variables were described as frequencies and percentages. Before performing the statistical analysis, the normality of distributions was evaluated to determine the most adequate test for each case. Chi-Square or Fisher’s exact test were used for comparisons between qualitative variables, and in the case of continuous ones, a *t*-test for two independent variables/ANOVA test (>2 categories) or Mann–Whitney/Kruskal–Wallis U test was performed. Time-to-event was analyzed with the Kaplan–Meier technique, estimating the median values and their IC95%, with comparisons among categories performed through the Log-rank test.

Statistical significance was established at *p* ≤ 0.05. All analyses were performed through the SPSS version 26.0 statistical software (SPSS, an IBM company, Chicago, IL, USA).

## 3. Results

During the study period, 380 infections were recorded and finally classified as 233 local (61.3%) and 147 systemic (38.7%) infections, of which 126 (33.2%) were referred from other centers (Figure 1). A high risk of infection before this episode was determined in 84 patients (22.1%), with 13.9% (*n* = 53) presenting two or more generator replacements, 22.9% (*n* = 87) undergoing an inspection or update of the device with the introduction of new leads, and 7.1% (*n* = 27) who had experienced a previous infectious episode. Overall, 133 (35.0%) cases were classified as acute, 117 (30.8%) as delayed, and 130 (34.2%) as late infections. Considering the time of onset, 182 episodes (47.9%) occurred after the first implant, and the rest after successive manipulations. The main characteristics of the patients and the risk categorization are summarized in Table 1. Microbial etiology is depicted in the Appendix A. *Staphylococcus epidermidis* and *Staphylococcus aureus* were the predominant microorganisms, with a greater prevalence of *S. aureus* in systemic infections and acute cases. Of note, 58 (15.3%) cases of systemic infection occurred more than a year after the implantation (late infections) and only the leads were affected. In nine of these cases produced by *S. aureus*, a distant source of the bacteremia could be identified: six were catheter-related bacteremias and three were wound infections after different types of surgery.

Vegetations were detected using echocardiography in 77/147 (52%) of patients with systemic infections, although transesophageal echocardiography was performed in only 98 patients. Vegetations were localized on the leads (*n* = 57) and over the leads and tricuspid valve (*n* = 20). A distant spread of the infection was diagnosed in patients with systemic infections: 29 (18%) patients developed pulmonary emboli that correlated with the presence of vegetations, 8 patients developed spondylitis, 3 patients developed septic arthritis (knee, sacroiliac, and sterno-clavicular joints), and 3 patients suffered from left-sided endocarditis.

### 3.1. Mortality and Costs for Local and Systemic Infections Depending on Final Classification (233 Local Infections and 147 Systemic Infections)

Local infections (*n* = 233) had an overall mortality of 2.5% (*n* = 6). Four cases were due to the extraction technique and the others were due to complications unrelated to the procedure during the early postoperative period. An additional two patients died due to other reasons during the first year of follow-up. The mortality rate was higher in systemic infections (10.8%, *n* = 16), mostly because of sepsis (Table 2).

Overall, the cost of infections was very high (median of EUR 21,790 for local and EUR 34,086 for systemic infections), in a great part attributed to hospital stays (7 ± 4.1 days and 24 ± 14.4 days, respectively), which represented 46% and 74% of total costs (Table 3). Costs also varied depending on the type of device implanted. ICD/ICD-RCT devices were the most expensive, as shown in Figure 2.

### 3.2. Outcome and Costs Regarding Local and Systemic Infections Depending on the Initial Therapeutic Approach

Of note, 48 of the 147 systemic infections (32.7%) were initially classified as local infections and progressed to systemic throughout the therapeutic process (four of them with fatal consequences). Thus, we found 281 local infections and 99 systemic infections at their initial presentation (Figure 1). The initial approach for local infections was often local surgery, which was usually unsuccessful. In fact, out of 281 initial local infections, 94 patients (33.5%), required more than one surgical procedure for definitive resolution. Multiple local procedures were performed in 42 cases (14.9%) and 50 (17.8%) of these patients were hospitalized for more than 24 h, with an overall success rate of only 20% (Table 4).

#### 3.2.1. Local Infections

Regarding the initial therapeutic approach, all patients with local infection who failed exclusive antimicrobial treatment (58.3%, *n* = 21) underwent TLEs. In 116 patients with an initial local approach, only 15 (12.9%) were successful. Of the remaining 111 patients, 4 were treated with chronic suppressive therapy, 2 progressed to systemic infection (1 died of sepsis and the other had the system removed by cardiac surgery after a failed TLE), and the other 2 died of unrelated causes. The remaining 97 patients (47 of whom developed systemic infection) underwent TLE, 2 of whom died as a result of the technique and 1 in the immediate postoperative period due to left ventricular failure. The average cost of this initial local approach was high (EUR 42,978) due to the repeated procedures, which almost always resulted in a final TLE or even cardiac surgery in cases that progressed to systemic infection (Table 5).

In contrast, performing TLE as the initial technique (*n* = 129) was more effective in these local infections, being successful in 121 patients (83.7%) at a lower cost (EUR 24,699), although three deaths related to this procedure were recorded.

Notably, in this group with local infections, TLE as the initial therapy or as a salvage procedure failed to achieve a complete extraction in 44/254 patients (14.4%) of which 7 (15.9%) experienced a subsequent relapse as a systemic infection. Six patients in this group underwent cardiac surgery, two of whom died, and one patient was given chronic suppressive antimicrobial treatment and died the following year of an unrelated cause (Appendix A).

#### 3.2.2. Systemic Infections

In the 99 patients who initially presented with systemic infections, antimicrobial treatment alone was associated with high mortality (19.2%, *n* = 5) and a cure rate of only 7.6% (*n* = 2). Two patients underwent open cardiac surgery and the remaining ninety patients (19 after failed antimicrobial therapy and 71 as the initial technique) underwent TLE, which cured 78 of them (86.6%) with 6 deaths (none related to the procedure). Incomplete extraction was observed in 13 cases (14.4%) and there were six recurrences (46.1%), with five subjects undergoing cardiac surgery for definitive resolution (Table 6 and Appendix A). Compared to antimicrobial treatment alone, TLE was associated with lower average costs (EUR 37,546 vs. EUR 39,525).

Out of a total of 337 TLEs, nine major complications (2.6%) were reported: four venous system ruptures (three deaths and one successful repair) and five myocardial tears (one death and four successful surgical repairs). Overall, 53 (15.7%) patients had incomplete TLEs (15 cases with residual intravascular fragment < 4 cm). The median age of the extracted leads was 5.7 (IQR: 1–10) years, with success rates of 89% (229/256) and 68.6% (54/81) in patients with leads <10 years and >10 years, respectively. A total of 49 TLEs were performed in octogenarian patients (32 local and 17 systemic infections), where this technique was the initial approach in 30 subjects and after another failed treatment, in 19 subjects, with successful results in all cases. However, clinical cure was achieved in 42 cases, with seven patients (14.2%) dying during hospitalization from various complications unrelated to the technique.

### 3.3. Reimplantation and Final Survival

In eighteen patients (4.7%), a complete withdrawal of the system was performed without reimplantation. A total of 287 patients received a new implant in another location. In 18 of these, the placement of a contralateral system was completed in another center prior to complete extraction and in 269, it was performed at our center. Of these 269 replacements, 199 (74.0%) were performed during the same extraction procedure and 70 (26.0%) were performed in two stages. Reinfection (all local) of the new implant during the one-year follow-up period was higher in the two-stage group (8/199, 4% vs. 7/70, 10%; *p* = 0.05), as was the length of hospital stay: 11 (IQR: 7–21) vs. 28 (IQR: 21–42) days; *p* < 0.01 (Table 7) and the costs: EUR 25,600 vs. EUR 44,797. This cost saving was partly due to the shorter hospital stay, but also to the fact that the group of patients with two-stage replacement had more expensive devices (see Appendix A). Of the 358 surviving patients in the entire cohort, 11 patients (3.1%) experienced a new infection during follow-up (coinciding with a new exchange or manipulation) and survived the new CIED infection. A total of 11 patients (3.1%) were lost to follow-up and 134 (37.4%) died of unrelated causes after a median follow-up of 4.75 (IQR: 1.6–9) years. No significant differences in late mortality were observed either according to the type of infection or the time of reimplantation (Appendix A).

## 4. Discussion

### 4.1. Mortality Rate Related with CIED Infections

According to the results of this study, mortality was considerable in the case of systemic infections (11%) even after successful TLE, mainly due to complications inherent to the surgical procedure in a frail and elderly population. These data are comparable to those published in a recent systematic review and meta-analysis that established an overall mortality rate of 13.7% [24], and highlight the need for establishing a very close multidisciplinary management of these patients.

### 4.2. Underwent TLE and Replacement

One of the concerns in performing TLE is the possibility of lead breakage in those with an implantation time of more than 10 years. For example, it should be noted that 17.3% of patients in this study had an incomplete extraction and in seven patients with an initial local infection, the process relapsed as a systemic infection. The progression of the infectious process from a local site (pocket) to the intravascular system can be considered a constant and should not be surprising. In fact, the classification of infection as local or systemic is largely based on clinical manifestations, as there is no definitive diagnostic test that allows the correct classification. In a study published by Klug et al. [25], of 105 long-standing local infections without contact with the infected pocket in which the leads were removed through a femoral approach, 79.3% of the tip cultures were positive, with similar values reported by our group [16]. It appears that the diffusion of microorganisms from the generator pocket infection through the wires into the vascular stream means that it is only a matter of time before clear clinical (fever, malaise), microbiological (positive blood cultures), or imaging (TEE with vegetations or wire uptake on positron emission tomography) manifestations are identified. Therefore, the incomplete removal of the entire system in local infections should alert us to a subsequent relapse as a systemic infection, sometimes difficult to recognize by non-expert clinicians, manifesting as fever without a clear infectious focus. In some cases, the retained fragments can be removed by femoral loops, but in others, the only solution is cardiac surgery, with a not inconsiderable mortality in frail patients. Chronic suppressive antimicrobial treatment has been proposed in stable patients [13,14], although recurrences are common after cessation of treatment, as we have shown in this report.

### 4.3. Cost Related with CIED Infections

There are several studies in the literature that refer to the costs of infections, although comparisons between countries are difficult given the wide variation in payment systems and their precise allocation in different healthcare systems. Average costs range from EUR 11,440 in Poland [26] to EUR 20,623–23,234 in France [27], GBP 14,241–30,958 in the UK [6,28] and USD 45,512–57,332 in the USA [1,5,29], in each case, twice the cost of conventional CIED implantation. However, many of these reports are based on data from national healthcare databases, where only total costs are calculated, sometimes without distinguishing between local and systemic infections, or differences due to different initial approaches. Our study reports a mean cost of EUR 21,790 for local infections, which rises to EUR 34,086 for systemic infections, mainly due to prolonged hospital stays, as has been pointed out in other studies [26,29,30]. In line with the data from our analysis, Egea et al. [31] reported total costs in our country that ranged from EUR 21,196 to EUR 41,496, depending on the type of device removed, being higher for ICD/CRT devices.

### 4.4. Cost-Effectiveness Measures

The antimicrobial treatment alone was associated with a high mortality (19.2%) and a poor healing rate (7.6%) in systemic infections. This study confirms that the most effective treatment for a CIED infection is the complete removal of the system. This approach is widely accepted in the case of systemic infections, but it is always accepted in local ones, usually due to its unclear identification (i.e., diagnosing many local infections as “sterile decubitus”), a lack of accessibility to a referral center for TLE, and the concerns regarding the performance of a complex surgical intervention on an elderly and fragile population with reduced survival expectations. However, advancements in medicine have resulted in greater life expectancy in this population, while increasing the need for these devices, and this trend will likely continue as the survival benefits of CIED demonstrated in clinical trials are translated to those observed in real-world practice. Also, this study showed that TLE had a lower average cost compared to antimicrobial-only treatment approach (EUR 37,546 vs. EUR 39,525).

In patients with a non-systemic CIED infection, local surgical techniques are usually employed, reporting good results in several studies [15,32,33,34]. However, most of these reports usually involve a short follow-up time with an evident publication bias, sometimes without providing either a description of all the difficulties attached to previous procedures or the associated costs. The present study demonstrates the efficiency of an aggressive initial approach, even in the oldest and most fragile patients, who are often subjected to multiple local techniques that lead almost inexorably to recurrences and continuous hospital stays that do not solve the problem and increase the final costs. In contrast, as has been also reflected in this cohort, the complete removal by TLE accounts for up to 2–4% of all major complications [10], some of them presenting high mortality. However, in expert hands, it is an effective technique and its performance in a surgical setting can resolve up to 50% of them and the mortality rate attributable to the technique is not exceeding 2% [35]. In addition, the cost is lower when it is considered as the initial approach. Moreover, lead fibrous adhesions to the veins are generally softer in older people [36,37,38], which makes their extraction easier, as we observed in this study with successful extractions in all our octogenarian patients. Therefore, according to the results, age should not be considered an absolute contraindication, as has also been demonstrated in other studies [39,40].

### 4.5. Replacement of the Device

Our study found interesting results regarding the frequent performance of implantation replacement in a single stage. This approach reduced the length of hospital stay (with cost savings) and optimized operating room use. In a survey performed in Europe, most centers implanted devices in a second stage with an interval ranging from 48 h to 2 weeks, depending on the infection type [41]. In this period, temporary pacemakers were inserted via jugular, femoral, or epicardial access. In fact, the clinical practice guidelines recommend device removal with a temporary pacing system and implantation of the new definitive system if the blood cultures are negative at 72 h [7,8,9]. However, the value of this “expert recommendation” for local infections has been questioned by several authors [36,37,38,39], showing that replacement in a one-stage intervention is not associated with a higher incidence of reinfections. In systemic infections, the reluctance to perform the replacement at the same time as the removal is even greater because it is assumed that replacement in the same intervention could contaminate the new system with the old. However, antimicrobial therapy strongly affects the adhesion of microorganisms to abiotic materials, even at suboptimal concentrations, as shown in in vitro studies [42] and experimental models [43] Thus, there is no convincing reason to believe that there is a higher risk of infection of the new device in a patient receiving appropriate antimicrobial therapy and with negative blood cultures who undergoes complete removal of the infected material. This has been shown in other implants [44] and, in fact, in our series, there was no association between one-stage replacement and a higher reinfection rate. In fact, when these procedures were split into two stages, there was a higher rate of infection in the new device by microorganisms resistant to the administered therapy. Long-term antimicrobial therapy might favor greater skin colonization by species naturally insensitive to the antibiotic, as was observed in other studies [45,46,47,48].

### 4.6. Prevention Measures

Given that removal of the entire CIED system may be necessary if infection occurs, prevention is of paramount importance to minimize the risk. In this study, up to 86 patients (22%) were considered to be at high risk of infection, and in 72 of these, the infection occurred in the following year, possibly acquired during the surgical procedure (primo-implant or manipulation). In 58 and 14 cases, these infections were local and systemic, respectively, resulting in three deaths and a total cost of EUR 2,197,687. Several measures have been advocated to prevent the development of infections in CIEDs, the most widely accepted being antimicrobial prophylaxis and avoidance of bleeding [49]. The implantation of antibiotic elution is currently a promising strategy to prevent CIED in high-risk patients. The commercially available absorbable antibacterial sheath TYRX™ (Medtronic Inc., Monmouth Junction, NK, US) has demonstrated a sustained 40–60% reduction in CIED infections in both observational and large randomized trials, particularly in high-risk patients [50,51,52,53].

However, despite the proven clinical benefit of this envelope, its use is associated with additional costs (estimated at EUR 990–1100 in Spain), which could lead to an additional financial burden on the healthcare system. Thorough cost-effectiveness analyses are therefore required for decision-making. These have been carried out by calculating the incremental cost-effectiveness ratio (ICER), usually expressed as the cost invested for the quality-adjusted life years (QALY) gained by implementing the new intervention compared with standard care. Studies based on the WRAP-IT population and incorporating the PADIT risk score identified willingness-to-pay thresholds of USD 112,603 per QALY in USA [52], EUR 40,000 in Italy, EUR 50,000 in Germany, and GBP 30,000 in England. Base case scenario analyses showed this product to be cost-effective, particularly in patients with ≥6 PADIT scores for all device types, assuming an overall infection rate of 1.9% [53].

However, in the Canadian healthcare system, this antimicrobial envelope has recently shown a lack of cost-effectiveness for any type of device in the base case scenario, except for higher infection rates (>6%) [54]. In Spain, a study [55] estimated a lower cost per QALY (EUR 22,000–25,000), and there is a lack of solid data on CIED infection rates due to the absence of a national registry and the small number of papers reporting these data. In our hospital, which performs more than 600 procedures per year, the infection rate over the last five years has ranged from 1.8% (PMs) to 2.8% (ICD/ICD-CRTs). In this scenario, TYRX™ is likely to be cost-effective, at least for high performance devices, and the introduction of an outcome-based risk-sharing programme from 2019 is likely to have a positive impact on costs.

## 5. Limitations

Finally, our study has some limitations. First, it is a long-term cohort performed by a single team that has gained experience over the years. Incomplete extractions have decreased over the years with increased experience and improved extraction tools, so the numbers should be viewed with caution. Secondly, our hospital has acted as a reference center, receiving most failed local approaches from other centers, so there may be some selection bias in this regard. It is possible that local approaches performed in other hospitals with good evolution (and therefore not referred) had a higher success rate than those observed here. However, a review of our own cases, including local approaches at the start of the programme, showed a similar number of failures. Thirdly, it should be noted that in most cases (74%), the placement of a new pacemaker after extraction was performed as part of a one-step approach, which allowed for a shorter length of stay. In other centers, the most common practice in cases of infection (especially if systemic) is to leave a variable period (between 2 and 4 weeks) until the placement of a new system, if it is performed. In PM-dependent patients, this period is usually spent in hospital, and often in the intensive care unit after general anesthesia, to ensure proper electrocardiographic monitoring for 24–48 h. This would have a greater impact on the final length of stay and therefore on the total cost, which could be much higher. On the contrary, the cost of devices in this study was calculated using prices in 2018. The increasing use of new devices has driven prices down, so it is possible that costs will be lower in the future.

## 6. Conclusions

In conclusion, the results of this study underline the importance of infections associated with CIEDs that are associated with increased patient morbidity and mortality, and represent a significant financial burden to the healthcare system. Local infections, far from being a minor problem, are frequently the precursor to systemic infection and we believe that they should be treated aggressively, always using TLE for complete removal of the system, which has been shown to be a cost-effective and safe strategy, even in fragile patients.

## Figures and Tables

**Figure 1 microorganisms-12-00537-f001:**
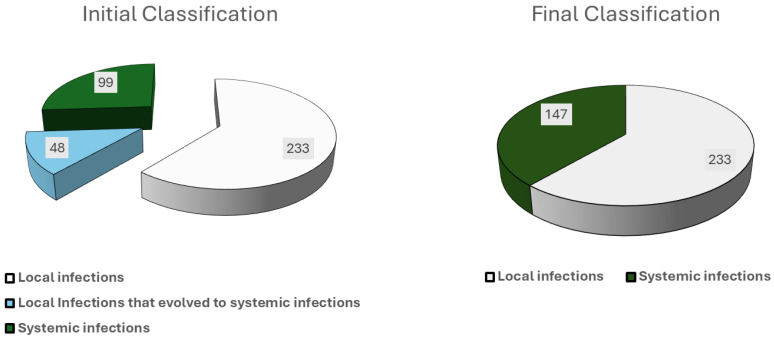
Initial classification (depending on its presentation) and final classification.

**Figure 2 microorganisms-12-00537-f002:**
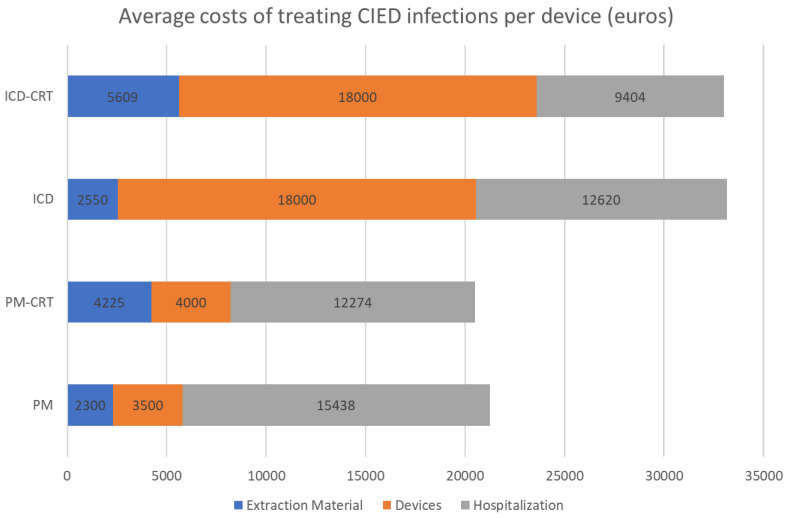
Average costs of treating CIED infections per device used (euros). CIED: cardiac implantable electronic devices; ICD-CRT: cardiac resynchronization therapy with defibrillator; ICD: implantable cardioverter-defibrillator; PM: pacemaker; PM-CRT: cardiac resynchronization therapy with pacemaker.

**Table 1 microorganisms-12-00537-t001:** Main characteristics of patients and risk of infection.

	*n*	%	PADIT Score
Male sex	283	74.5	
Age			
<60 years	82	21.6	2
60–69 years	93	24.5	1
>69 years	205	53.9	0
Implant type			
PM	264	69.5	-
ICD	73	19.2	-
PM-CRT	8	2.1	-
ICD-CRT	35	9.2	-
Procedure			
First PM implantation	179	47.1	0
First ICD implantation	148	39.0	2
CRT	53	14.0	4
Revision/Update	87	22.9	4
Previous procedures			
None	179	47.1	0
One	148	38.9	1
Two or more	53	13.9	3
Kidney failure	45	11.8	1
Immunosuppression	21	5.5	1
Previous infection	27	7.1	-
Other comorbidities			
Arterial Hypertension	193	50.8	-
Diabetes mellitus	132	34.7	-
Ischemic cardiopathy	133	35	-
Myocardiopathy	107	28.2	-
COPD	68	17.9	-
Stroke	21	5.5	-
Neoplasia	18	4.7	-
Liver cirrhosis	5	1.3	-
Two comorbidities	62	16.3	-
Three comorbidities	75	19.7	-
More than three	72	18.9	-
Initial classification ^1^ Local Infection Systemic infection	281 99	74 26	
Final Classification ^2^ Local infection Systemic infection	233 147	61.3 38.7	

^1^: In this classification, infections are grouped according to their initial presentation. ^2^: In this classification, infections are grouped according to their final diagnosis. A total of 48 infections presented as local but evolved to systemic and are grouped together here with the 99 systemic ones that originally presented as such. CRT: cardiac resynchronization therapy; ICD: implantable cardioverter defibrillator; PM: pacemaker; COPD: chronic obstructive pulmonary disease.

**Table 2 microorganisms-12-00537-t002:** Causes of mortality (one year) depending on the type of infection.

Local Infections	
Reasons	*n*
Related to the technique: - Rupture of superior vena cava - Cardiac tamponade - Ventricular arrhythmia during TLE	4 2 1 1
Related to the infection: - Ventricular failure post-TLE - Stroke post-TLE	2 1 1
Non-related to the technique or the infection: - Acute leukemia - Left ventricular failure	2 1 1
**Systemic infections**	
**Reasons**	*n*
Related to the technique: - Rupture of superior vena cava	1 1
Related to the infection: - Septic shock without performed extraction *. - Septic shock with performed extraction *. - Stroke post-TLE - Nosocomial pneumonia post-TLE - Ventricular failure post-TLE - Left-sided infective endocarditis **	15 5 4 2 1 1 1
Non-related to the technique or the infection: - Traffic accident	1

*: In 8 cases, the etiology was *S. aureus* and in the other a polymicrobial infection by *S. aureus* and *P. aeruginosa.* **: Left-sided infective endocarditis by *Pseudomonas aeruginosa.*

**Table 3 microorganisms-12-00537-t003:** Overall and broken-down costs (expressed in euros) of CIED infections (final classification).

	Median	CI 95%	Range
**Local infections (*n* = 233)**			
Extraction material	3600	2550–4738	0–11,950
Devices	3500	3500–7000	0–126,000
Hospital stay/care	10,068	9128–11,180	476–110,160
Overall cost	21,790	19,842–23,398	476–152,165
**Systemic infections (*n* = 147)**			
Extraction material	2300	0–3700	0–11,950
Devices	3500	3500–7000	0–72,000
Hospital stay/care	25,360	22,068–28,840	5704–171,177
Overall cost	34,086	30,800–37,116	3980–247,927

**Table 4 microorganisms-12-00537-t004:** Progression of the 281 local infections (initial classification) according to the initial therapeutic approach performed at their initial presentation.

	Deaths *n* (%)	Failures *n* (%)	Healing *n* (%)	Contralateral Recurrence *n* (%)	Overall *n*
Local Surgery					
Exclusive antimicrobial treatment ^1^	0 (0.0)	21 (58.3)	15 (41.7)	-	36
Pocket debridement ^2^	0 (0.0)	44 (74.6)	15 (25.4)	-	59
Generator replacement and alternative system ^2^	0 (0.0)	15 (100.0)	0 (0.0)	0 (0.0)	15
Various procedures ^2^	0 (0.0)	42 (100.0)	0 (0.0)	-	42
Transvenous Lead extraction					
Initial TLE ^3^	5 (3.8)	3 (2.3)	4 (3.1)	117 (90.6)	129
Secondary TLE ^4^	3 (2.4)	4 (3.2)	3 (2.4)	115 (92)	125
Total TLE	8 (3.1) ^5^	7 (2.7) ^5^	232 (91.3)	7 (2.7) ^6^	254

TLE: transvenous lead extraction. ^1^ Out of the 21 patients with failure, in 2, a local surgery was performed (with a new failure) followed by a final successful TLE (with complete healing), and in the remaining 19, TLE was performed with complete healing but contralateral recurrence in 3, in which a new successful extraction was performed. ^2^ Out of 101 patients, from these 116 with local surgical approaches (59 + 15 + 42) recurred after local surgery (one or more procedures): in 4, a chronic antimicrobial treatment was administered. Two of them progressed to systemic infections, with one dying and another one, after an unsuccessful TLE, underwent cardiac surgery and was finally cured. The other two died during the follow-up due to unrelated causes to the infection. The remaining 97 patients were subjected to TLE, registering two deaths due to the technique and one death due to acute left ventricular failure after extraction. ^3^ These patients initially underwent TLE as the initial therapeutic approach. ^4^ These patients underwent TLE after failed to other approaches. ^5^ Five patients died due to the technique (three for rupture of the superior vena, one from cardiac tamponade, and one from ventricular fibrillation during the extraction) and three during the immediate postoperative period due to non-related complications (one for a stroke and two for ventricular failure). ^5^ In 44 (17.3%) of 254 TLEs, the tracking was incomplete. The seven patients who relapsed after an incomplete extraction developed a systemic infection and six went to cardiac surgery, two of them died and the other one was in chronic suppressive antimicrobial therapy. ^6^ Four patients who developed an infection of the new implant underwent a new successful TLE, two healed with antimicrobial therapy and, in the remainder, a local approach (surgical debridement) was performed, with curation.

**Table 5 microorganisms-12-00537-t005:** Overall and broken-down cost (expressed in euros) of infection in cardiac electrostimulation devices, according to their initial diagnosis (initial classification) and the first therapeutical approach ^1^.

	*n*	Overall Cost, Mean (SD)	Extraction Material, Mean (SD)	Devices, Mean (SD)	Hospital Stay/Care, Mean (SD)
**Local infections**					
Exclusive antimicrobial treatment	36	26,660 (22,584)	1982 (2641)	7638 (11,640)	17,038 (14,303)
Local surgery	116	42,978 (3570)	2629 (3256)	12,524 (19,188)	27,834 (29,237)
TLE	129	24,699 (16,374)	3187 (2841)	7046 (7263)	14,464 (13,826)
Overall	281	31,932 (27,408)	2811 (3000)	9215 (13,770)	19,906 (21,962)
**Systemic infections**					
Exclusive antimicrobial treatment	26	39,525 (32,606)	1823 (2245)	2758 (3522)	34,951 (33,192)
TLE	71	37,546 (30,227)	3148 (2184)	6542 (11,162)	27,856 (23,094)
Overall	97 ^2^	38,077 (30,722)	27,793 (1754)	5525 (9845)	29,758 (26,187)

TLE: transvenous lead extraction; SD: standard deviation. ^1^ After initial approach, in the cases of relapse/no healing, subsequent costs until complete resolution of the CIED infection (i.e., TLE or open-heart surgery) are added to each case. ^2^ Data not included from 2 patients who underwent initial cardiac surgery.

**Table 6 microorganisms-12-00537-t006:** Development of the 99 systemic infections (initial diagnosis) according to the initial and final therapeutic approach (note that 19 patients who failed with antimicrobial therapy are then included in the TLE group).

	Death *n* (%)	Failures *n* (%)	Healing *n* (%)	Contralateral Relapse ^1^ *n* (%)	Overall *n*
Exclusive antimicrobial treatment ^2^	5 (19.2)	19 (73.1)	2 (7.7)	-	26
TLE ^3^					
Complete	5 (6.5)	0 (0.0)	68 (88.3)	4 (5.2)	77
Incomplete	1 (7.7)	6 (46.2)	5 (38.5)	1 (7.7)	13
Cardiac surgery ^4^	0 (0.0)	0 (0.0)	6 (85.7)	1 (14.3)	7

TLE: transvenous lead extraction. ^1^ All cases with relapse with infection in the contralateral system underwent a new TLE and healed. ^2^ All deaths were due to septic shock due to an *S. aureus* infection. ^3^ Performed initially in 71 patients and as a second option in 19 patients in which exclusive antimicrobial treatment failed (90 patients altogether). Six deaths were unrelated to the technique, but to diverse complications (cerebrovascular accident, idiopathic ventricular fibrillation, sepsis, nosocomial pneumonia, etc.) occurring during the immediate postoperative period. Out of the six patients who presented with relapse after an incomplete TLE, five underwent cardiac surgery and were finally cured. In the other patient, a chronic suppressive antimicrobial treatment was administered and they subsequently died. ^4^ Cardiac surgery was performed in 7 patients: 5 after failed TLE and as the first approach in 2 other patients in which TLE was judged extremely difficult.

**Table 7 microorganisms-12-00537-t007:** Characteristics of the patients who underwent transvenous lead extraction (TLE) and replacement in one or two stages.

Local Infections			
	One Step (*n* = 133)	Two Steps (*n* = 45)	*p*
Type of device PM ICD/CRT	115 (90%) 13 (10%)	7 (16%) 36 (84%)	<0.01
Hospitalization stay (days)	8 (5, 13)	9 (5, 17)	NS
Duration ATB treatment (days)	21 (16, 28)	21 (21, 30)	NS
Replacement interval (days)	0	6 (4, 9)	
Nº of relapses (%)	3 (2.2%) #	1 (2.2%) ##	NS
Nº of reinfections (%)	2 (1.5%) ***	1 (2.2%) ****	NS
Systemic Infections			
	One Stage (*n* = 75)	Two Stages (*n* = 28)	*p*
Type of device PM ICD/CRT	71 (94.6%) 4 (5.3%)	10 (35.7%) 18 (64.2%)	<0.01
Hospitalization stay (days)	22 (16, 32)	32 (10, 53)	<0.05
Duration ATB treatment (days)	29 (28, 45)	32 (21, 43)	NS
Replacement interval (days)	0	11 (7, 25)	
Nº of relapses (%)	1 (1.3%) *	1 (3.5) **	NS
Nº of reinfections (%)	2 (2.6%) &	4 (14.2%) &&	0.05

PM: pacemaker; CRT: cardiac resynchronization therapy; ICD: implantable cardioverter defibrillator; ATB: antibiotic. #: *S. epidermidis* (3) ##: *S. epidermidis*. ***: *C. acnes* → *S. epidermidis*; *S. lugdunensis* → *Corynebacterium striatum*. ****: *S. epidermidis* MS → *S. epidermidis* MR. *: *P. aeruginosa*, **: *S. aureus*. &: *S. hominis* MS → *S. epidermidis* MR; *S. aureus* → *P. aeruginosa*. && *S. aureus* → *P. aeruginosa*; *S. aureus* MS → *S. schleiferi* MR; *S. aureus*/*S. epidermidis* → *Fusobacterium nucleatum*; *S. aureus* MS → *S. epidermidis* MR; NS = not significant.

## Data Availability

Data are contained within the article and Appendix A.

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
