# Peer review of "Mortality and Costs of Cardiac Implantable Electronic Device (CIED) Infections According to the Therapeutic Approach: A Single-Center Cohort Study"

_microorganisms, 2024, doi:10.3390/microorganisms12030537_

Round 1

Reviewer 1 Report (Previous Reviewer 2)

Comments and Suggestions for Authors

As a reviewer of your article titled “Mortality and Cost Related with Cardiac Implantable Electronic Devices (CIED) Infections according to the Therapeutic Approach: A Single-Centre Cohort Study,” I find your work to be insightful and well-structured. However, I suggest a few significant modifications to enhance the clarity and depth of your study:

1. Abstract Revision: The abstract requires an additional comparative element to contextualize the term "high." It is unclear what the mortality rates or costs are being compared to. Additionally, it's essential to specify whether the findings are statistically significant, aligning with the statistical benchmarks set in your methods section.

2. Flow Chart for Classification: The article would benefit from a flow chart detailing the study outline, especially focusing on the initial and final classifications of infections. This visual aid will help clarify cases where infections initially classified as local later evolved into systemic infections. Such a progression could indicate a more severe outcome due to potential under-diagnosis, which is a critical aspect of your study.

3. Pathogens and Source Details: The article mentions microbiological procedures in the supplemental material but lacks specific details about the pathogens involved and their sources. Including this information will provide a more comprehensive understanding of the infections associated with CIEDs, which is vital for the study's context and relevance.

4. Statistical Evidence for Comparison: When discussing the high mortality in cases of systemic infections, it is imperative to provide statistical evidence to back this claim. This also applies to the cost assessment of the infections. Without statistical validation, these claims may lack the necessary rigor and could be questioned by the scientific community.

Incorporating these suggestions will significantly strengthen your article by providing clearer comparisons, better visual representation of the study design, more detailed information on pathogens, and necessary statistical evidence to support your claims.

Comments on the Quality of English Language

Please check the previous comments on the quality of English Language.

Author Response

Reviewer 2 Report (Previous Reviewer 1)

Comments and Suggestions for Authors

I find that previous reviewer comments were adequately addressed and the quality of the paper has significantly increased. Well done!

Author Response

Reviewer 3 Report (New Reviewer)

Comments and Suggestions for Authors

Though retrospective and single-centre, the study analysed data from a large number of patients with CIED infections of current interest in daily practice.

Major concerns:

...............................

- In the methods, the definition of CIED infection type should be clarified more. Was it based on the authors' definition or available literature data (consensus, guidelines)? Patients with isolated pocket signs whose extracted lead cultures are positive: are they still considered "local infection"?

Same for temporal definition (acute, delayed, late), any reference?

Residual leads < 4 cm without impacting clinical outcomes is considered "clinical success". How did the authors classify such patients, and why?

- The long period of the included patients (nearly three decades) is a major limitation. Practising TLE and available technologies before 2000 is completely different from recent times.

- The study period is clear, but the median follow-up is not. For example, mortality refers to in-hospital, during one-year follow-up. please clarify

- Systemic infection increased by about 50% from initial to final classification. Was this dependent on the treatment approach? This is an important point to clarify since cost analysis may vary if the initial classification is considered. What is the author's rationale for considering only the final classification?

- Mortality rate (a part of the article title) should also be mentioned in systemic infection within the abstract. Moreover, the mortality rate was low in the non-extraction group; was this due to mortality related directly to TLE procedures? This might give the wrong message that a conservative approach improves survival?!.

- How the authors explain the higher re-infection rates in patients undergoing TLE and reimplantation in 2 steps (leaving an adequate period to heal from infection) vs the one-step approach. One may expect the opposite.

- Calculating the cost of procedures performed in 1985 based on device cost in 2018 is confusing. Any statistical way to homogenize the cost analysis over this long study period (starting in 1985!).

Minor comments:

....................................

- Vegetation may impact the TLE approach and outcomes; any data?

- No mention of possible bridge therapy in pacing-dependent patients or shifting to other devices (leadless PM, S-ICD) and the possible impact on clinical outcomes and related costs.

- In literature, about 1/4 of patients who underwent TLE did not undergo reimplantation for the lack of indication. What is the authors' experience in this regard? 

Author Response

Reviewer 4 Report (New Reviewer)

Comments and Suggestions for Authors

This is an interesting study which describes the costs and the outcomes of a Spanish cohort of patients with CIED-infection. This study is a valuable work which underlines the economic burden of the CIED-infection. However, I believe there are certain aspects of the study that could be further improved and clarified for improved comprehension."

Line 75: Please clarify the inclusion and exclusion criteria of your study. A flow-chart of the inclusion process that you followed for this study would be useful

Line 80: I understand that you have collected prospectively the data. I would like to know who collected the data (the clinician who treated the patient? Infectious disease residents trained to the protocol? Other kinds of researchers?) and who have assessed the patients’ outcome. In consideration of the extensive duration of this study, it is presumed that data has been collected from various sources. Please clarify the specific origins of the data, such as whether it was derived from electronic medical records or any other designated sources. Furthermore, I think that, at least the part of data collection should be clearly reported in the manuscript for a better comprehension of the readers.

Line 83: I think that it is important to report at least the number of endocarditis and not only the distinction between local and systemic infections

Line 155: I do not understand the classification of this cohort of patients, who have a confirmed diagnosis of CIED-infection, using a score typically employed to assess the risk of infection in patients with a CIED and a suspicion of infection. In the Discussion section, it is mentioned that 72 patients, initially considered at high risk of infection, subsequently experienced an infection. However, the timing of the PADIT score calculation for these patients remains unclear. Could you please provide clarification in the Methods section, specifying whether the PADIT score was calculated before or after the diagnosis of CIED-infection?

Line 165: delete “While”

Line 200: Please clarify in the Methods what you mean with “sepsis”. A  “Definitions” section would be useful for a better comprehension of the readers.

Line 228-Table 3: It is not clear to me what you mean with “Local surgery” here. In the lines 119-121 you defined the local surgical approach including in this approach the “generator extraction and placement of an alternative system”, while in Table 3 you have classified it separately. Please clarify what kind of procedure these 59 patients classified as “Local surgery” went through.

Table 6: How much patients who received a replacement procedure died during the follow-up period? Please report the differences in terms of mortality between the one step and two step groups

Line 394: “its cost is lower” instead of “it cost is less”

Line 395-396: Please provide a reference supporting this phrase (“lead fibrous adhesions to the veins are generally softer in older, which makes its extraction easier”

Line 397-398: I would like to acknowledge the critical role of the operator's expertise in influencing the success of a procedure. Your group works in a referral centre for the CIED-infection. It is plausible that this success rate observed in your octogenarian patients is at least partially attributable to your expertise in this field. Please reformulate this phrase in a more polite way.

Line 399: “demonstrated by these two works” instead of “demonstrated by the evidence”

Line 427: This phrase is not clear (“except for higher infection rates (>6%)”).

Line 431: Which scenario?

I would like to suggest that this paragraph (“Prevention measures”) may include some redundant information and may slightly deviate from the primary focus of your work. Specifically, there are references to CIED-infection prevention techniques not directly related to your study. A brief reference emphasizing the significance and cost-effectiveness of the prevention measures would be sufficient.

Line 443: This statement may be considered redundant unless accompanied by the reporting of outcomes from your own cases

Lines 444-451: Do you have the data regarding the overall costs of the one-step procedure compared to the two-step procedure group?

Maybe you should underline that both the European and the American guidelines (cite the last update published on 2023) suggest to delay reimplantation until signs and symptoms of local and systemic infection have resolved. Furthermore, it would be right to provide the current evidence which support the one-step procedure. Here you can find some suggestions:

-        Mountantonakis SE, Tschabrunn CM, Deyell MW, et al. Same-day contralateral implantation of a permanent device after lead extraction for isolated pocket infection. Europace 2014;16:252–7.

-        Chew D, Somayaji R, Conly J, et al. Timing of device reimplantation and reinfection rates following cardiac implantable electronic device infection: a systematic review and meta-analysis. BMJ Open 2019;9:e029537. doi:10.1136/ bmjopen-2019-029537

-        Tascini C, Giuliano S, Attanasio V, Segreti L, Ripoli A, Sbrana F, Severino S, Sordelli C, Weisz SH, Zanus-Fortes A, Leanza GM, Carannante N, Di Cori A, Bongiorni MG, Zucchelli G, De Vivo S. Safety and Efficacy of a Single Procedure of Extraction and Reimplantation of Infected Cardiovascular Implantable Electronic Device (CIED) in Comparison with Deferral Timing: An Observational Retrospective Multicentric Study. Antibiotics (Basel). 2023 Jun 2;12(6):1001. doi: 10.3390/antibiotics12061001. PMID: 37370320; PMCID: PMC10295375.

I would like to highlight the significance of the majority of your patients undergoing a one-step procedure, as this aspect is crucial in supporting the relevance of your work. Maybe you should underline that this unconventional approach could be an interesting strategy to reduce the costs related to the CIED-infection management

Line 460: It would be better to be more polite and to align with the results of your work (e.g. “The results of this study underline the importance of…”)

Comments on the Quality of English Language

minor revision

Author Response

Reviewer 5 Report (New Reviewer)

Comments and Suggestions for Authors

This manuscript aims to analyze the treatment methods and costs of local/systemic infections in patients with CIED placement, in order to determine which treatment is effective for the two groups of infections and to minimize the costs incurred in treatment. The innovation of this article is generally acceptable, but the logic in the results section is confusing, and the discussion is not comprehensive.

1. For clinical analysis, a multi-center study with a larger number of cases is more convincing.

2. The results section has confusing logic. Section 3.1 describes the costs and mortality rates of treatment based on local/systemic infection grouping, but the article does not include a graphical analysis of mortality rates in this part; only costs are analyzed. Section 3.2 analyzes costs and clinical prognosis based on initial grouping, but the cost situation is not presented. It is also unclear whether grouping based on initial infection type is meaningful. Should a consistent grouping method be adopted for analysis throughout the article? Sections 3.3 and 3.4 have confusing titles and are not well-aligned with the analysis and charts in the text.

3. The discussion section does not explore which treatment method yields the most benefits based on infection type groupinglocal or systemic.

4. Sections 4.4 and 4.5 discussed in the discussion part seem unrelated to the main content of the article.

5. Tables in the article could be more intuitive and better understood.

Comments on the Quality of English Language

Minor editing of English language required

Round 2

Reviewer 1 Report (Previous Reviewer 2)

Comments and Suggestions for Authors

Almost all responses were reasonable, except of "Mortality is considered as 'high' because in medicine, a mortality greater than 10% is always considered 'high,' and there is no need for explanation" is not entirely accurate. While a mortality rate exceeding 10% often raises significant concern, it requires further context and nuanced interpretation, even in the abstract.

Author Response

We changed the sentence in the last version (including the abstract). Please, note that we have discussed our results with a meta-analysis of the literature (ref 24) in the discussion.

Reviewer 5 Report (New Reviewer)

Comments and Suggestions for Authors

The revised manuscript is more logically organized, with thorough discussions. It would be clearer if pie or column charts  were used more frequently in the graphical representations instead of the ternary diagrams. 

Comments on the Quality of English Language

English narration of the manuscript is fine.

Author Response

Ok. A new figure (pie) is provided in the new version.

This manuscript is a resubmission of an earlier submission. The following is a list of the peer review reports and author responses from that submission.

Round 1

Reviewer 1 Report

Comments and Suggestions for Authors

Extensive English proofreading is mandatory.

The scope of the article is limited.

The manuscript should be improved in terms of  in terms of clarity and focus.

Clearly state what this study adds to the existing knowledge. Why is this important to the readers? What are the implications for this field of knowledge?

I believe the paper would be much more interesting if it did not focus on the costs, but on the outcomes of these patients.

The discussions section could be improved by providing in-depth analysis and insights based on the information presented in the results section.

The study only has 31 references.

Comments on the Quality of English Language

Quality of English must be improved. 

Reviewer 2 Report

Comments and Suggestions for Authors

General Comments

=============

I had the opportunity to peer-review your article titled “Outcomes and costs of Cardiac Implantable Electro-stimulation Devices (CIED) according to the therapeutic approach.” The article is well-written and presents valuable insights. However, I recommend the following modifications for further clarity and depth.

Specific Comments

=============

Major Comments

---------------------

1. Title Clarity: The title should be more specific regarding CIED-related infections. A clearer title will enhance the focus and scope of the article.

2. Abstract - Patient Background: Include age and sex of the cases in the abstract. This information is crucial for readers to quickly understand the demographic details of the study population.

3. Introduction - Previous Literature: The introduction needs elaboration. Specifically, incorporate findings from previous articles about the outcomes and costs of CIED. It's advisable to shift some parts of the results section (particularly lines 281-366) to the introduction for better context setting.

4. Methodology - Study Location Details: Add a brief description of the study's location, such as the size of the hospital and the annual number of procedures performed. This information will provide readers with a better understanding of the study setting.

5. Infection Classification Methodology: Clarify the criteria used to decide whether an infection is systematic or local. This distinction is critical for understanding the study's methodology and results.

6. PADIT Score Explanation: Introduce or briefly explain the PADIT score, as it seems to be a significant element in your study but lacks sufficient introductory information.

7. Pathogens and Source Details: If available, include details about the pathogens involved and their sources. This is particularly relevant considering the mention of microbiological procedures in the supplemental material.

8. Mortality Comparison Clarification: When stating that mortality was high in cases of systemic infections, specify what this is compared to (e.g., general population) and provide statistical evidence to support this claim.

9. Discussion Section Structure: It is recommended to use subheadings in the discussion part. Directly discuss the results in relation to previous literature and/or general population for a more organized and coherent analysis. 

These modifications will significantly enhance the clarity, depth, and scientific rigor of your article.

Comments on the Quality of English Language

Please check the previous "Quality of English Language"